# Antibiotic Resistance and Presence of Persister Cells in the Biofilm-like Environments in *Streptococcus agalactiae*

**DOI:** 10.3390/antibiotics13111014

**Published:** 2024-10-28

**Authors:** Pamella Silva Lannes-Costa, Isabelle Rodrigues Fernandes, João Matheus Sobral Pena, Brunno Renato Farias Verçoza Costa, Marcel Menezes Lyra da Cunha, Bernadete Teixeira Ferreira-Carvalho, Prescilla Emy Nagao

**Affiliations:** 1Laboratory of Molecular Biology and Physiology of Streptococci, Institute of Biology Roberto Alcantara Gomes, Rio de Janeiro State University—UERJ, Rio de Janeiro 20550-013, RJ, Brazil; pamella.lannes@uerj.br (P.S.L.-C.); fernandes.isabelle@graduacao.uerj.br (I.R.F.); pena.joao@posgraduacao.uerj.br (J.M.S.P.); 2Núcleo Multidisciplinar de Pesquisa UFRJ—Xerém em Biologia, Campus UFRJ—Duque de Caxias Professor Geraldo Cidade, Universidade Federal do Rio de Janeiro, Rio de Janeiro 25240-005, RJ, Brazil; brunno.vercoza@caxias.ufr.br (B.R.F.V.C.); marcel@caxias.ufrj.br (M.M.L.d.C.); 3Departamento de Microbiologia Médica, Universidade Federal do Rio de Janeiro, Rio de Janeiro 21941-902, RJ, Brazil; bernadete@micro.ufrj.br

**Keywords:** *Streptococcus agalactiae*, persister, antibiotic resistance

## Abstract

**Objectives:** This study investigated antibiotic resistance and presence of persister cells in *Streptococcus agalactiae* strains belonging to capsular types Ia/ST-103, III/ST-17, and V/ST-26 in biofilm-like environments. **Results:**
*S. agalactiae* strains were susceptible to penicillin, clindamycin, and erythromycin. Resistance genes were associated with *tet*M (80%), *tet*O (20%), *erm*B (80%), and *lin*B (40%). Persister cells were detected in bacterial strains exposed to high concentrations of penicillin, clindamycin, and erythromycin. *S. agalactiae* capsular type III/ST-17 exhibited the highest percentage of persister cells in response to penicillin and clindamycin, while type Ia/ST-103 presented the lowest percentages of persister cells for all antimicrobials tested. Additionally, persister cells were also detected at lower levels for erythromycin, regardless of capsular type or sequence type. Further, all *S. agalactiae* isolates presented efflux pump activity in ethidium bromide-refractory cell assays. LIVE/DEAD fluorescence microscopy confirmed the presence of >85% viable persister cells after antibiotic treatment. **Conclusions:** These findings suggest that persister cells play a key role in the persistence of *S. agalactiae* during antibiotic therapy, interfering with the treatment of invasive infections. Monitoring persister formation is crucial for developing strategies to combat recurrent infections caused by this pathogen.

## 1. Introduction

*Streptococcus agalactiae* is the leading cause of invasive infections, such as bacteremia and meningitis, in neonates and adults with comorbidities [1,2]. The pathogen is a commensal colonizer of the respiratory, gastrointestinal, and female genital tracts. The vertical transmission of *S. agalactiae* from mother to baby during labor is considered the source of early-onset neonatal disease (EOD), occurring within the first 6 days of life [3,4]. Thus, screening pregnant women for *S. agalactiae* colonization and intrapartum antibiotic prophylaxis has been successful in reducing EOD [5]. However, these prevention strategies have had no effect on late-onset disease (LOD) in infants aged 7–90 days [6,7]. In recent years, *S. agalactiae* has also gained global recognition in non-pregnant adults, particularly in the population with comorbidities such as diabetes, cancer, obesity, and hypertension [8,9].

Recently, reduced susceptibility to penicillin and resistance to macrolide and fluoroquinolone have been described in *S. agalactiae* [10]. Resistance or decreased sensitivity to penicillin may be due to the production of β-lactamases, mutations in penicillin-binding proteins (PBPs), or variations in molecular interactions as altered vibrational modes that weaken the stability of the penicillin-PBP complex, allowing bacteria to continue cell wall synthesis despite the presence of the drug [11]. A genetically identical bacterial population, called persister cells, can present phenotypic diversity that allows a small fraction of the population to survive the bactericidal effect of antibiotics [12,13,14]. Although persisters are genetically identical to the subpopulation sensitive to antimicrobials, they are able to enter a physiologically dormant state for survival [12]. Therefore, in a population containing persister cells, the death kinetics will be biphasic during treatment with antimicrobials, where the initial steep slope will reflect the death of the sensitive part of the population, followed by a slower death rate, corresponding to the fraction of persisters, until reaching a plateau. Persister cells have been proposed as a major cause of recurrence of several chronic infections, as well as an increased risk of emergence of antibiotic resistance [15]. Persisters have previously been described in *Escherichia coli*, *Streptococcus pyogenes*, *Pseudomonas aeruginosa*, *Salmonella typhimurium,* and *Listeria monocytogenes* [16,17,18].

Stress response, efflux pumps, inhibition of ATP production, toxin–antitoxin systems, reactive oxygen species, and cellular communication are some suggested mechanisms for persister cell formation. However, the destruction of persister cells is still challenging, as their formation remains unknown [19,20]. Understanding the mechanisms of formation of antibiotic-persistent cells may aid in the development of new therapeutic approaches for the treatment of chronic and recurrent bacterial infections.

This study revealed the presence of >85% viable persister cells in *S. agalactiae* clusters (biofilm-like environment) in response to penicillin, clindamycin, and erythromycin, suggesting that persistence is a general feature in the *S. agalactiae* population.

## 2. Results and Discussion

The *S. agalactiae* COH1-III, GBS90356-III, and GBS1428-III strains belong to sequence type 17 (ST-17). GBS85147-Ia belongs to ST-103, and CNCTC10/84-V to ST-26 were used in this study. The *S. agalactiae* ST-17 hypervirulent clone is associated with neonatal invasive infections, and antimicrobial resistance rates require particular attention [2,21,22]. Research from Taiwan showed that ST-17 often harbors resistance genes, contributing to its persistence and spread [23]. The prevalence of ST-17 and its association with increased risks of invasive infections in newborns make it a key target for global surveillance efforts and vaccine development [24]. Therefore, continued surveillance is needed to monitor the spread of ST-17, especially given the emergence of multidrug-resistant strains.

ST-103 was described as an environmental transmission clone, important for *S. agalactiae* dissemination [25]. *S. agalactiae* strains belonging to ST-103 have been identified in neonatal sepsis isolates [26], asymptomatic pregnant women [27], and cows in Norway [28]) and China [29]. ST103 was also found to be the most common sequence type in bovine milk in Denmark [30] and China [31]. Additionally, ST-103 has been identified among human *S. agalactiae* isolates in developing economies [32], indicating that the potential transmission of this genotype between bovines and humans, or vice versa, cannot be ruled out [24]. Strains of *S. agalactiae* ST-26 have also been identified in asymptomatic adults and neonates with invasive diseases in Japan, USA, UK, and Sweden [33,34,35,36], demonstrating the pathogenic potential of this clone in human infections.

Antibiotic therapy is a primary option for the treatment of invasive infections in humans. Consequently, antibiotic resistance has been the subject of extensive research worldwide [37]. According to the globally adopted intrapartum antibiotic prophylaxis procedure, pregnant women colonized by *S. agalactiae* are treated with penicillin or erythromycin if they are allergic to penicillin. However, this prophylaxis reduces the incidence of early-onset diseases, but does not demonstrate a significant reduction in late-onset diseases, causing high neonatal morbidity and mortality [38]. In the present study, the characterization of the antimicrobial susceptibility of *S. agalactiae* showed that all *S. agalactiae* strains were susceptible to penicillin, clindamycin, erythromycin, vancomycin, levofloxacin, azithromycin, chloramphenicol, ceftriaxone, and linezolid. These findings align with previous published reports [10,39,40,41], indicating that penicillin remains the preferred first-line antibiotic, with erythromycin and clindamycin as alternative treatments for individuals with penicillin allergies. However, high resistance to tetracycline (60%) was observed (Table 1). Although tetracyclines are not the first-line treatment for *S. agalactiae* infections, recent studies suggest that increasing resistance to tetracyclines in *S. agalactiae* may be a side effect of treatments for other bacterial infections [42,43]. High tetracycline resistance rates have also been observed in bovine *S. agalactiae* in China [44], as well as in humans in Brazil [10,45] and in Saudi Arabia [46]. Furthermore, more than half of the strains analyzed in China exhibited tetracycline resistance [47].

These data highlight the critical need for continuous monitoring of antibiotic resistance and raise concerns about the use of antibiotics for non-therapeutic purposes. Therefore, conducting antimicrobial resistance studies is essential to establish rational and effective guidelines for antimicrobial therapy, reducing the risk of developing and spreading antimicrobial resistance [48].

In our study, resistance to tetracycline genes was associated with *tet*M 80% (n = 4), and only strain GBS1428-III presented the *tet*O gene (20%; n = 1). In addition, *erm*B (80%; n = 4) and *lin*B (40%; n = 2), which are responsible for resistance to macrolides and lincosamides, were also verified. Our results are in agreement with the publication of [49] that analyzed *S. agalactiae* strains isolated from non-pregnant, pregnant, and neonate populations worldwide between 2000 up to 2021, showing *tet*M as the leading antibiotic resistance gene, followed by *erm*B. *tet*M was also the main tetracycline resistance gene detected in elderly patients with respiratory disease in China [50], pregnant women and newborns in Brazil [10], and colonized and infected adults in Saudi Arabia [46]. It is of concern that phenotypically susceptible bacterial strains may harbor antimicrobial resistance genes. In the case of erythromycin, prevalent in vitro susceptible strains harbored the *erm*B gene [37].

Presently, the *erm*A gene (encoding erythromycin resistance) was not detected among *S. agalactiae* strains (Table 1), corroborating data obtained in Taiwan [51] and Tehran, Iran [52]. Moreover, among the 115 strains studied in China, only one presented the *erm*A gene [43]. In our data, the presence of the *mef*AE gene, which encodes macrolide efflux gene, was observed only in the GBS1428-III strain (Table 1). Although other publications state that the *mef*A gene is the most common in *S. agalactiae* ST-17 strains [26], GBS90356 and COH1 strains, both ST-17, did not present the *mef*AE gene. The *mef* gene is often associated with macrolide resistance in *S. agalactiae*. However, its role in virulence is less clear. Although the presence of *mef* genes contributes to antibiotic resistance, specifically to macrolides such as erythromycin, their involvement in increased virulence has not been definitively proven. Some studies indicate that strains of *S. agalactiae* carrying the *mef* gene exhibit higher survival rates under antibiotic pressure, potentially allowing these strains to persist longer in host environments [53]. Further research is required to determine if the *mef* gene indirectly contributes to virulence by enabling the persistence and spread of antibiotic-resistant strains, leading to more severe disease [21].

Interestingly, *S. agalactiae* strains that presented the *lin*B, *mef*A, and *erm*B genes did not show resistance to lincosamides or erythromycin in the antimicrobial susceptibility test. Our data can be explained by the fact that specific signals or stressors are responsible for triggering the expression of resistance genes, which can lead to transient phenotypic resistance [54]. Genotypic–phenotypic correlation analysis has shown that the presence of the resistance gene may not be detected phenotypically in an unstimulated environment [55]. However, the genotypic–phenotypic correlation of antimicrobial resistance in *S. agalactiae* is poorly understood, requiring future studies to optimize the treatment and public health impact of invasive infections caused by this microorganism.

The minimum inhibitory concentration (MIC) test was performed on penicillin (MIC ≤ 0.03 μg/mL), clindamycin (MIC ≤ 0.25 μg/mL), and erythromycin (MIC ≤ 0.25 μg/mL). MIC values for penicillin (MIC range = 0.015–0.03 μg/mL), clindamycin (MIC range = 0.03–0.06 μg/mL), and erythromycin (MIC range = 0.12–0.25 μg/mL) were demonstrated in Table 1. Persister cells were detected for *S. agalactiae* strains in all antibiotic concentrations tested, MICs of 8 μg/mL for penicillin, 6 μg/mL for clindamycin, and 4 μg/mL for erythromycin. The MIC value cannot be determined for persister cells, because by progressively increasing the drug concentration, using the same high population density inoculum, the MIC value cannot be detected. In addition, when regenerated in the absence of the antimicrobial, the persisters completely restore drug susceptibility to MIC values.

Persister cells have been observed during treatment with several classes of antibiotics. This phenomenon represents a form of persistence induced not only by antibiotic exposure but also by other stressors such as starvation, extreme pH levels, temperature changes, and DNA damage [16]. However, spontaneous persistence can also arise due to individual variations in cellular molecular and biochemical processes that affect gene expression, resulting in the persister state in a more random manner [13]. Stress-triggered or spontaneous persister cells are temporary; however, they can complicate the treatment of bacterial infections by causing persistence and recurrence of infections [18].

Currently, to evaluate persister cell formation in *S. agalactiae*, bacterial strains were inoculated onto agar base plates with and without supplementation of 5% defibrinated sheep blood. The plates, containing 8 μg/mL penicillin, 6 μg/mL clindamycin, and 4 μg/mL erythromycin, were covered with cellophane membranes or left uncovered. Persister cells were formed when high bacterial loads (10^10^ CFU/mL) were inoculated. The mean detection of persister cells of *S. agalactiae* cultured in medium containing 8 μg/mL penicillin (MIC ≥ 0.015 µg/mL) corresponded to 2.3% (*p* = 0.0010) of the total population of bacterial cells cultured in the absence of penicillin (3.2–2.7 × 10^10^ CFU/mL) for COH1-III, while for GBS90356-III and GBS1428-III (MIC ≥ 0.03 µg/mL, each), persister cells corresponded to 1.03% each (*p* = 0.0017 and *p* = 0.0384, respectively) of the total population of cells cultured in the absence of antibiotic (8.0–6.0 × 10^10^ CFU/mL and 4.8–3.2 × 10^10^ CFU/mL, respectively). The CNCTC10/84-V strain showed 0.37% (*p* = 0.0268; MIC ≥ 0.03 µg/mL) of persister cells in relation to the total population cultured in the absence of the drug (5.6–4.0 × 10^10^ CFU/mL), and the GBS85147-Ia strains showed 0.27% (*p* = 0.0390) of persister cells (MIC ≥ 0.03 µg/mL) in relation to the total cell population cultured in the absence of the drug (8.0–5.3 × 10^10^ CFU/mL) (Figure 1A; Table 1).

Similar to the results observed with penicillin, persister cells were also detected at concentrations of 6 μg/mL clindamycin and 4 μg/mL erythromycin. The mean percentage of persisters cells for clindamycin was 5.06% (*p* = 0.0030) to COH1-III (MIC ≥ 0.03 µg/mL), while for GBS90356-III (MIC ≥ 0.0625 µg/mL), the persister cells corresponded to 3.43% (*p* = 0.0210), followed by 3.02% (*p* = 0.0414) for GBS1428-III (MIC ≥ 0.03 µg/mL), 2.76% (*p* = 0.02816) for CNCTC10/84-V (MIC ≥ 0.0625 µg/mL), and 1.97% (*p* = 0.0405) for GBS85147-Ia (MIC ≥ 0.0625 µg/mL). The total population of cells cultured in the absence of drugs was the same as previously described for each strain (Figure 1A; Table 1).

The lowest percentage of persister cells was observed for erythromycin. The persister cells for the COH1-III strain was 1.77% (*p* = 0.0028; MIC ≥ 0.125 µg/mL), followed by CNCTC10/84-V with 1.66% (*p* = 0.0275; MIC ≥ 0.25 µg/mL), GBS90356-III with 1.49% (*p* = 0.0203; MIC ≥ 0.25 µg/mL), GBS1428-III with 1.44% (*p* = 0.0387; MIC ≥ 0.25 µg/mL), and GBS85147-Ia with 1.11% (*p* = 0.0399; MIC ≥ 0.25 µg/mL) of persister cells (Figure 1A; Table 1). Persisters on uncovered BAB plates formed small hemolytic colonies that returned to their normal size after being transferred to fresh drug-free media. Phenotypic reversion was observed when persister cells were grown without antibiotics or in non-biofilm-like environments, indicating the involvement of non-inherited antimicrobial resistance mechanisms. Studies have demonstrated the effect of biofilm and high cell density on the failure of antibiotics to eliminate microorganisms such as mycobacteria and methicillin-resistant Staphylococcus aureus [56,57]. Similarly, clindamycin-induced persister Streptococcus pyogenes cells were detected only in cells grown in biofilm-like environments [16]. Furthermore, Pseudomonas aeruginosa persister cells were observed under biofilm conditions, suggesting that the agglomerated cell environment contributed to the persistent phenotype of *P*. aeruginosa to tobramycin [58]. Our results are in agreement with previously described data, providing valuable insight for persisters formation during treatment of *S. agalactiae* infections.

Efflux pumps are key mechanisms involved in bacterial antibiotic resistance, and their role in persister cell formation has attracted attention [13,19]. In this study, an EtBr efflux assay was performed to assess efflux pump activity for *S. agalactiae* strains during the persister formation. The MIC value of all *S. agalactiae* strains for EtBr was 0.12 µg/mL. However, bacterial growth was observed at the highest concentration of EtBR used to allow persister formation (4 μg/mL), demonstrating the participation of the efflux pump. The percentage of EtBr-refractory cells recovered at a concentration of 4 μg/mL to COH1-III was about 0.88% (*p* = 0.0020) of *S. agalactiae* population grown in biofilm-like environments in the absence of EtBr (7.0–6.4 × 10^10^ CFU/mL), while for GBS90356-III, it was 0.67% (*p* = 0.0189) of the bacterial population grown in the absence of EtBr (3.3–2.5 × 10^10^ CFU/mL), followed by GBS1428-III with 0.64% (*p* = 0.0135) of the *S. agalactiae* population grown in the absence of EtBr (6.7–5.3 × 10^10^ CFU/mL). The percentage of EtBr-refractory cells recovered from CNCTC10/84-V and GBS85147-Ia was 0.49% (*p* = 0.0006) and 0.28% (*p* = 0.0113) of the bacterial population grown in biofilm-like environments in the absence of EtBr (8.0–7.6 × 10^10^; 9.3–7.5 × 10^10^ CFU/mL), respectively. All strains showed efflux pump activity, through to EtBr-refractory cells (Figure 1B; Table 1).

Previous results described by Greve et al. (2024) [18] suggested that the efflux pump mechanism was essential for the survival of *S. agalactiae* persister cells, which may explain the persistence of this bacterium during antibiotic treatment. Other research has also linked efflux pumps to the survival of bacteria under antibiotic stress [16,19]. S. pyogenes persister cells induced in biofilm-like environments exhibited increased EtBr efflux activity, suggesting the involvement of efflux pumps in their antibiotic tolerance [16]. Furthermore, the authors suggested that efflux pump activity would be related to cell growth arrest, reinforcing the idea that persisters use multiple strategies to survive in hostile conditions. Therefore, understanding the role of efflux pumps during bacterial persistence will contribute to new treatment strategies to combat recurrent infections.

These data were corroborated by LIVE/DEAD stain using the apotome scanning microscopy (Figure 2). Only *S. agalactiae* COH1-III, GBS90356, and GBS85147 strains were analyzed, since they represented the highest and lowest percentages of persister cells verified in the present study. Fluorescence analysis of persister cells allowed distinguishing between live and dead cells. Single cells, small aggregates, and accumulations of cells could be visualized regularly. A high concentration of live cells was observed when bacterial cells were not treated with antibiotics (94–98% controls). *S. agalactiae* strains treated with a high concentration of penicillin exhibited viability ranging from 85 to 94%. Additionally, the quantification of viable bacteria for clindamycin treatment ranged from 90 to 92%, while for high concentration of erythromycin, viable cells ranged from 89 to 91% (Appendix A). Our data demonstrate for the first time the visualization of persister cells after treatment with multiple antimicrobials.

In summary, this study demonstrated the presence of persistent antibiotic-tolerant *S. agalactiae* cells in response to penicillin, clindamycin, and erythromycin, regardless of capsular type or sequence type, which may explain recurrent disease after treatment of invasive infections.

## 3. Materials and Methods

### 3.1. Bacterial Culture Conditions and Identification Procedures

*S. agalactiae* cultures were stored at −80 °C in Brain Heart Infusion liquid aliquots (BHI; Difco Laboratories, Detroit, MI) with 20% glycerol. *S. agalactiae* isolates were cultured on blood agar base (BAB; Oxoid, Cambridge, UK) plates containing 5% sheep defibrinated blood for 24 h at 37 °C. *S. agalactiae* strains were screened by the following phenotypic conventional tests: β-hemolysis on BAB, Gram-positive cocci, negative catalase reaction, and CAMP-test positive. The isolated bacteria were serologically confirmed as group B Streptococcus of the Lancefield group, using a commercial streptococcal grouping kit (DR0584A Oxoid, Brazil), according to the recommendations of the manufacturer [59].

### 3.2. Capsular Typing of S. agalactiae Strains

*S. agalactiae* (COH1 isolated from a case of human neonatal sepsis; CNCTC10/84 isolated from blood of a septic neonate; GBS90356 isolated from a 3-day-old neonate with fatal meningitis; GBS1428 isolated from urine of a 71-year-old man with colorectal carcinoma; and GBS85147 isolated from the oropharynx of an adult patient; n = 5) were subjected to multiplex PCR assays. The final volume of each reaction mixture was 50 μL, containing 200 μM dNTPs, 2.5 units GoTaq polymerase (Promega Biotechnology, São Paulo, Brazil), 50 ng DNA, and 10 pMol of reverse and forward primers for the respective capsular types in a thermal cycler (Veriti; Applied Biological, New York, NY, USA). The cycling conditions were as follows: denaturation at 96 °C for 2 min by 35 cycles, annealing at the respective annealing temperature for 1 min and extension at 72 °C for 30 seg, with a final elongation step of 72 °C for 2 min followed by a hold at 4 °C. Amplification was verified in a 1.2% agarose gel stained with Sybr Green and thereafter visualized and photographed under UV light. Methodology submitted for patent registration BR102017018169-3.

### 3.3. Antimicrobial Susceptibility

*S. agalactiae* strains (n = 5) were assayed by the disk diffusion methods using Mueller Hinton agar plates, and antimicrobial disks were obtained commercially (BD Diagnostic Systems, São Paulo, SP, Brazil). The antimicrobials tested were penicillin 10 μg (PEN), clindamycin 2 μg (CLIN), erythromycin 15 μg (ERY), vancomycin 30 μg (VAN), levofloxacin 5 μg (LEV), tetracycline 30 μg (TET), azithromycin 15 µg (AZI), chloramphenicol 30 µg (CHL), ceftriaxone 30 μg (CEF), and linezolid 30 μg (LIN) [60].

### 3.4. Detection of Antimicrobial Resistance Genes

The antibiotic resistance genes (*erm*A, *erm*B, *mef*AE, *lin*B, *tet*M, and *tet*O; Appendix A) of *S. agalactiae* strains (COH1, CNCTC10/84, GBS90356, GBS1428, and GBS85147) were detected by PCR. Macrolides and lincosamides are the antibiotic alternative for patients with a history of β-lactam allergy. However, the increasing resistance among clinical strains of *S. agalactiae* [61] to macrolides and lincosamides, as well as the ubiquitously high (usually > 80%) resistance to tetracycline in *S. agalactiae* [41], highlight the importance of these resistance genes. Then, the PCR amplification reaction was performed in a 50 µL mixture volume composed of 200 μM each dNTP, 2.5 units of GoTaq polymerase (Promega Biotechnology, Brazil), 50 ng DNA, and 10 pMol of reverse and forward primers in a thermal cycler (Veriti; Applied Biological, New York, NY, USA). Amplification of resistance genes was as follows: pre-denaturation at 94 °C for 5 min, denaturation at 94 °C for 1 min, annealing at 52 °C for 1 min, extension at 72 °C for 2 min, and final extension at 72 °C for 5 min. PCR products were resolved by electrophoresis in accordance with item 3.2 (Appendix A).

### 3.5. Minimal Inhibitory Concentration (MIC)

MICs for *S. agalactiae* strains (n = 5) were tested by the agar dilution method, as recommended by the Clinical & Laboratory Standards Institute [56], in concentrations ranging from 0.015 to 8 μg/mL penicillin (PEN) (Sigma, St. Louis, MO, USA), 0.03–6 μg/mL clindamycin (CLIN; Sigma), 0.12 to 4 μg/mL erythromycin (ERY; Sigma), and 0.015 to 4 μg/mL ethidium bromide (EtBr; Sigma). The assay was performed in biological triplicates and repeated at two technical replicates.

### 3.6. Identification of S. agalactiae Persister Cells

The study to generate persisters was performed as previously described by Martini and colleagues (2021) [16] for *Streptococcus pyogenes* strains. To simulate the bacterial agglomeration observed in biofilms, a cellophane membrane was placed on agar media containing antibiotics, allowing the formation of a bacterial film on its smooth surface after inoculating a high bacterial load (biofilm-like environment). To prepare the bacterial inoculum, *S. agalactiae* isolates (COH1, CNCTC10/84, GBS90356, GBS1428, GBS85147) were grown in BHI containing 0.5% (*w*/*v*) of yeast extract (BHI-Y) at 37 °C for 4 h to reach the exponential phase. After centrifugation, the pellet was adjusted (~1–2 × 10^10^ colony forming unit-CFU/mL) using the same broth. To form a bacterial film, a 100 μL volume was homogeneously spread on the surface of a cellophane membrane placed onto BHI-Y agar containing 5% defibrinated sheep blood (BAB) and supplemented with different antimicrobials for 18 h at 37 °C. Persisters can grow at antibiotic levels well above the MIC. Thus, concentrations of 8 μg/mL penicillin (PEN; Sigma), 6 μg/mL clindamycin (CLI; Sigma), and 4 μg/mL erythromycin (ERY; Sigma), which correspond to 16- to 267-fold higher levels than the MIC for each drug, were randomly selected in the present study. Persister cells were removed from the cellophane membranes at the highest drug concentration in which growth was detected for CFU counting [62]. To test whether defibrinated sheep blood affects the analysis, the experiments were also performed without blood. Antimicrobial-susceptible control cells were obtained exactly as described above but using inoculum size adjusted to concentrations recommended by CLSI (~10^6^ CFU/plate; condition that does not allow the generation of persisters). For each antimicrobial tested, three biological experiments were performed with two technical replicates each. Two CFU determinations were carried out for each dilution (n = 4).

### 3.7. Detection of Refractory Cells to Ethidium Bromide

The identification of efflux-competent bacterial strains can be determined sensitively and specifically by measuring increased EtBr MIC values [16,63]. Therefore, we evaluated the occurrence of EtBr-refractory cells in the biofilm-like environment for *S. agalactiae* strains (COH1, CNCTC10/84, GBS90356, GBS1428, GBS85147). A biofilm-like environment (~10^10^ CFU/plate) was applied onto cellophane membranes and incubated at 37 °C for 18 h. The cellophane membranes were examined for the presence of *S. agalactiae* cells at the highest concentration of EtBr where growth was observed, in order to determine the CFU. Controls were performed exactly as above but with susceptible cells (~10^6^ CFU/plate). Four biological experiments were performed with two technical replicates each. Two CFU determinations were carried out for each dilution (n = 4).

### 3.8. Fluorescence Staining Method

To examine the arrangement of viable and non-viable bacteria within a biofilm-like environment, LIVE/DEAD BacLight kit (Invitrogen Molecular Probes in Eugene, OR, USA) was utilized. The kit includes SYTO9, a green fluorescent stain for live bacteria, and propidium iodide, a red fluorescent stain for dead bacteria. The stains were combined in equal proportions (1:1) and mixed with sterile water at a ratio of 3 μL of the stain mixture per 1 mL of sterile water to create a staining solution [59]. The resulting stain solution was applied to the biofilm-like environment of *S. agalactiae* strains. The distribution of live bacteria, indicated by green fluorescence, and dead bacteria, indicated by red fluorescence, was subsequently observed using a fluorescence microscope (Zeiss Axio Observer 7—Apotome 3). The images were processed using the Fiji distribution of ImageJ [64]. Preprocessing steps for the LIVE/DEAD macro included background subtraction with a radius of 5.0 pixels. The parameters used for the LIVE/DEAD plugin were set to a prominence value of 15 for live cells and 25 for dead cells to avoid the inclusion of weak and/or diffuse staining. Additionally, a median filter with a radius of 0.9 pixels was applied. Automatic quantification of fluorescence-imaged live/dead assays was conducted in Fiji, following the protocol by Kerkhoff and Ludwig (2024) [65].

### 3.9. Statistical Analyses

To evaluate the quantity of persister cells recovered in the presence of β-lactams and other antimicrobial classes, including EtBr-refractory cells, a Student’s t-test and one-way ANOVA were conducted, followed by Tukey’s post hoc test for multiple comparisons [95% CI (confidence interval); * *p* < 0.05; ** *p* < 0.01; *** *p* < 0.001; **** *p* < 0.0001] [16].

## 4. Conclusions

In this study, the antimicrobial susceptibility profile revealed that although *S. agalactiae* strains (ST-103, ST-17, ST-26) were susceptible to the most antimicrobials tested, persistent cells to penicillin, clindamycin, and erythromycin were identified. Persister cells indicate an additional layer of complexity in the treatment of *S. agalactiae* infections. The mechanisms underlying this persistence, including efflux pump activity, suggest that persister cells may contribute to chronic infections and treatment failure. Therefore, knowledge of persistent bacterial populations is crucial to improve therapeutic approaches and reduce morbidity/mortality associated with *S. agalactiae* infections, especially in neonates, the elderly, and adults with comorbidities.

## Figures and Tables

**Figure 1 antibiotics-13-01014-f001:**
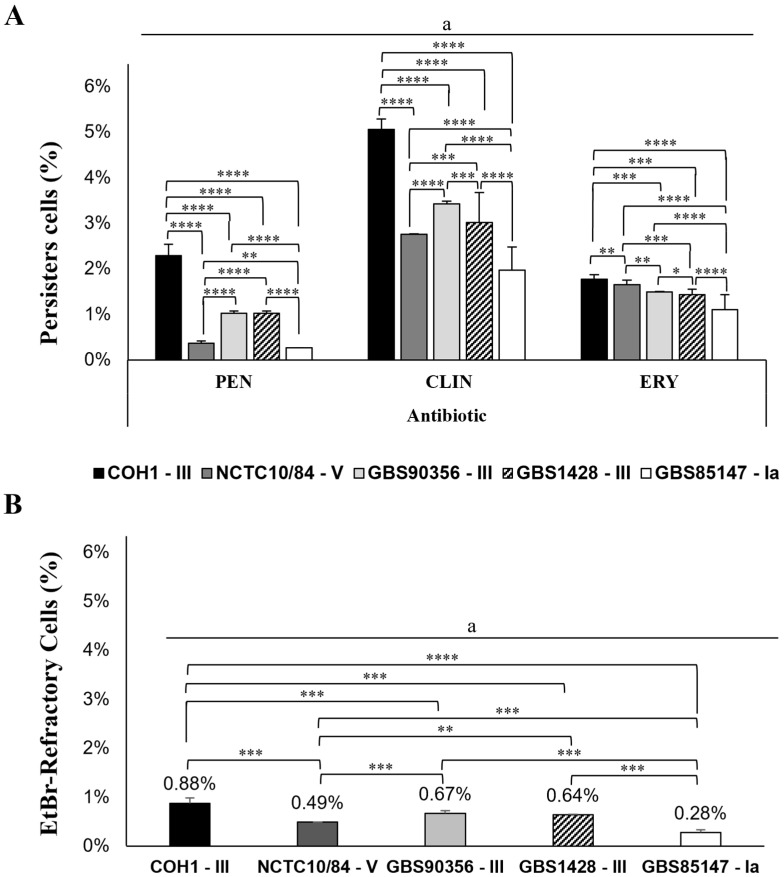
(**A**) *Streptococcus agalactiae* (COH1, CNCTC10/84, GBS90356, GBS1428, and GBS85147) persisters recovered from biofilm-like environments at concentrations of 8 μg/mL penicillin (PEN), 6 μg/mL clindamycin (CLI), and 4 μg/mL erythromycin (ERY); (**B**) EtBr-refractory cells recovered at concentration of 4 μg/mL; the average CFU/mL of the control cells (no antibiotic) corresponded to 100%. Comparison control and antibiotics ª *p* < 0.05; * *p* < 0.05; ** *p* < 0.01; *** *p* < 0.001; **** *p* < 0.0001.

**Figure 2 antibiotics-13-01014-f002:**
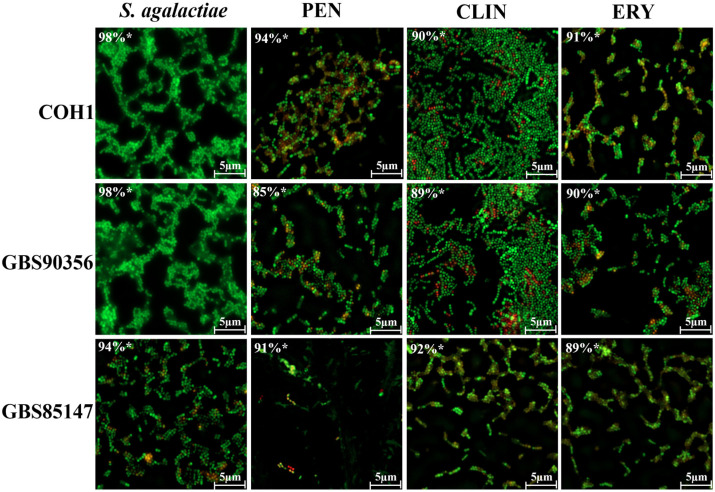
Fluorescence analysis of *S. agalactiae* persister cells. *Streptococcus agalactiae* (COH1, GBS90356, and GBS85147) control cells (no antibiotic) and persister cells at concentrations of 8 μg/mL penicillin (PEN), 6 μg/mL clindamycin (CLI), and 4 μg/mL erythromycin (ERY); (*) white asterisk represents the percentage of viable cells by LIVE/DEAD marking.

**Table 1 antibiotics-13-01014-t001:** Characterization of susceptibility antimicrobial of *S. agalactiae*.

*S. agalactiae*—Capsular Type	Antibiotics										Resistance Genes						MIC Values (µg/mL)			
	VAN	CLIN	LEV	PEN	TET	AZI	CHL	ERY	CEF	LIN	*ermA*	*ermB*	*mefAE*	*linB*	*tetM*	*tetO*	PEN	CLIN	ERY	EtBr
COH1—III	S	S	S	S	R	S	S	S	S	S	−	+	−	+	+	−	≥0.015	≥0.03	≥0.125	≥0.12
CNCTC10/84—V	S	S	S	S	S	S	S	S	S	S	−	+	−	−	+	−	≥0.03	≥0.0625	≥0.25	≥0.12
GBS90356—III	S	S	S	S	R	S	S	S	S	S	−	+	−	+	+	−	≥0.03	≥0.0625	≥0.25	≥0.12
GBS1428—III	S	S	S	S	R	S	S	S	S	S	−	−	+	−	−	+	≥0.03	≥0.03	≥0.25	≥0.12
GBS85147—Ia	S	S	S	S	S	S	S	S	S	S	−	+	−	−	+	−	≥0.03	≥0.0625	≥0.25	≥0.12

Legend: Breakpoints according to CLSI 2024 guidelines. Minimum inhibitory concentration (MIC); penicillin (PEN); clindamycin (CLIN); erythromycin (ERY); ethidium bromide (EtBr); vancomycin (VAN); levofloxacin (LEV); tetracycline (TET); azithromycin (AZI); chloramphenicol (CHL); ceftriaxone (CEF); linezolid (LIN); positive amplification (+); negative amplification (−); sensitive (S); resistant (R).

## Data Availability

The original contributions presented in the study are included in the article. Further inquiries can be directed at the corresponding author.

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
