# Peer review of "Antibiotic Resistance and Presence of Persister Cells in the Biofilm-like Environments in *Streptococcus agalactiae"

_antibiotics, 2024, doi:10.3390/antibiotics13111014_

Round 1
Reviewer 1 Report
Comments and Suggestions for Authors
The manuscript describes the investigation of the antibiotic resistance and persistence induced by Streptococcus agalactiae strains belonging to capsular types Ia/ST-103, III/ST-17, and V/ST-26 in biofilm-like environments susceptible to penicillin, clindamycin, and erythromycin.
The topic and scope fit the journal Antibiotics, and special issue “Streptococcus spp. and Enterococcus spp. in Humans and Animals: Virulence Potential, Antimicrobial Resistance, Genomic Trends and Approaches,” the methods are appropriate and some impact may be expected. English usage needs to be improved.
The reproducibility and rigor can be further improved: although the methods are described with details except for a few key steps, such as identification of the species (Section 3.1), multiplex PCR (3.2), PCR (3.4). As these methods are paramount to the execution of the eexperiment, details should be given rather than referring to the literature, patent database, etc. Section 3.9, the applied confidence intervals and alpha values have not been given.
The result dissemination is relatively well done.
Fig. 2 scale bars are not readable.
What is the pixel size?
The ordering of the 3 capsular types should be consistent throughout, in Abstract and Conclusion.
In the introduction, the work would be advantaged by further providing the fundamental molecular chemical aspects explaining the mechanism of drug resistance to penicillin and such, suggest citing Mucsi et. al., Phys. Chem. Chem. Phys, 15 (2013), 20447-20455.
The title indicates “induced by biofilm-like environments,” and does this mean the results are environment-sensitive?
Finally, the supplementary material, since it contains only a small data table it can be incorporated into the main manuscript; or if the authors prefer to have it as a separate file, the title should be “Supplementary information for Title” to avoid confusion.
Comments on the Quality of English LanguageModerate editing, Abstract needs to be re-written.
Author Response
Reviewer #1 comments:
The manuscript describes the investigation of the antibiotic resistance and persistence induced by Streptococcus agalactiae strains belonging to capsular types Ia/ST-103, III/ST-17, and V/ST-26 in biofilm-like environments susceptible to penicillin, clindamycin, and erythromycin.
The topic and scope fit the journal Antibiotics, and special issue “Streptococcus spp. and Enterococcus spp. in Humans and Animals: Virulence Potential, Antimicrobial Resistance, Genomic Trends and Approaches,” the methods are appropriate and some impact may be expected. English usage needs to be improved.
- The reproducibility and rigor can be further improved: although the methods are described with details except for a few key steps, such as identification of the species (Section 3.1), multiplex PCR (3.2), PCR (3.4). As these methods are paramount to the execution of the experiment, details should be given rather than referring to the literature, patent database, etc. Section 3.9, the applied confidence intervals and alpha values have not been given.
Lines 272-275 on pages 7: We wrote the sentences to read: The isolated bacteria were serologically confirmed as group B Streptococcus of the Lancefield group, using a commercial streptococcal grouping kit (DR0584A Oxoid, Brazil), according to the recommendations of the manufacturer [59].
Line 280-288 on page 7, 8: We wrote the sentences to read: …were subjected to multiplex PCR assays targeting nine cps (Ia, Ib-IX). The final volume of each reaction mixture was 50 μL containing 200 μM dNTPs, 2.5 units GoTaq polymerase (Promega Biotechnology, Brazil), 50 ng DNA, and 10 pMol of reverse and forward primers for the respective capsular types in a thermal cycler (Veriti, Applied Bio). The cycling conditions were as follows: denaturation at 96 °C for 2 min by 35 cycles, annealing at the respective annealing temperature for 1 min and extension at 72 °C for 30 seg with a final elongation step of 72 °C for 2 min followed by a hold at 4 °C. Amplification was verified in a 1.2 % agarose gel stained with Sybr Green and thereafter visualized and photographed under UV light.
Line 299-312 on page 8: The antibiotic resistance genes (ermA, ermB, mefAE, linB, tetM, and tetO; Supplemantary Materials Table S2) of S. agalactiae strains (COH1, CNCTC10/84, GBS90356, GBS1428, and GBS85147) were detected by PCR. Macrolides and lincosamides are the antibiotic alternative for patients with a history of β-lactam allergy. However, the increasing resistance among clinical strains of S. agalactiae [62] to macrolides and lincosamides, as well as the ubiquitously high (usually >80%) resistance to tetracycline in S. agalactiae [63], highlights the importance of these resistance genes. Then, the PCR amplification reaction was performed in a final 50 µL mixture volume comprised of 200 μM each dNTP, 2.5 units of GoTaq polymerase (Promega Biotechnology, Brazil), 50 ng DNA and 10 pMol of reverse and forward primers in a thermal cycler (Veriti, Applied Bio). Amplification of resistance genes was as follows: pre-denaturation at 94 °C for 5 min, denaturation at 94 °C for 1 min, annealing at 52 °C for 1 min, extension at 72 °C for 2 min and final extension ate 72 °C for 5 min. PCR products were resolved by electrophoresis in accordance with item 3.2 (Supplemantary Materials Figure S1).
Lines 378 on page 9 - We wrote the sentences to read: [95% CI (confidence interval); *p<0.05; **p<0.01; ***p<0.001; ****p<0.0001].
- The result dissemination is relatively well done. Fig. 2 scale bars are not readable. What is the pixel size?
Answer: The scale bars have been revised and the size adjusted to 600 dpi.
- The ordering of the 3 capsular types should be consistent throughout, in Abstract and Conclusion.
Answer: The ordering of the 3 capsular types was corrected throughout the Abstract and Conclusion.
- In the introduction, the work would be advantaged by further providing the fundamental molecular chemical aspects explaining the mechanism of drug resistance to penicillin and such, suggest citing Mucsi et. al., Phys. Chem. Chem. Phys, 15 (2013), 20447-20455.
Lines 43-47 on pages 1, 2 - We wrote the sentences to read: Resistance or decreased sensitivity to penicillin may be due to the production of β-lactamases, mutations in penicillin-binding proteins (PBPs), or variations in molecular interactions as altered vibrational modes that weaken the stability of the penicillin-PBP complex, allowing bacteria to continue cell wall synthesis despite the presence of the drug [11].
- The title indicates “induced by biofilm-like environments,” and does this mean the results are environment-sensitive?
Answer: Yes, persister cells formation is sensitive to the environment and can be reversed when persisters are grown without antibiotics or in non-biofilm-like environments. However, accepting the suggestion of reviewer 2, the title of the manuscript will be changed to "Antibiotic resistance and presence of persister cells in the biofilm-like environments in Streptococcus agalactiae”.
Line 204 -215 on page 6 - We wrote the sentences to read: Phenotypic reversion was observed when persister cells are grown without antibiotics or in non-biofilm-like environments, indicating the involvement of non-inherited antimicrobial resistance mechanisms. Studies have demonstrated the effect of biofilm and high cell density on the failure of antibiotics to eliminate microorganisms such mycobacteria and methicillin-resistant Staphylococcus aureus [56,57]. Similarly, clindamycin-induced persister Streptococcus pyogenes cells were detected only in cells grown in biofilm-like environments [16]. Furthermore, Pseudomonas aeruginosa persister cells were observed under biofilm conditions, suggesting that the agglomerated cell environment contributed to the persistent phenotype of P. aeruginosa to tobramycin [58]. Our results are in agreement with previously described data, providing valuable insight for persisters formation during treatment of S. agalactiae infections.
- Finally, the supplementary material, since it contains only a small data table it can be incorporated into the main manuscript; or if the authors prefer to have it as a separate file, the title should be “Supplementary information for Title” to avoid confusion.
Answer: The supplementary material will be attached as a separate file with the title “Supplemental information for the title”.
Reviewer 2 Report
Comments and Suggestions for Authors
The manuscript submitted by Lannes-Costa studied the antibiotic resistance and persistence in in Streptococcus agalactiae. This topic is interesting and the study is meaningful. In my opinion, there are some issues to be addressed:
1. There lacks solid data for the statement of “persistence induced by biofilm-like environments” in the title. In my opinion, the present data can only supported the presence of persister cells in the biofilm-like environment.
2. In the result section, how to interpret the result that there was linB gene, mefA and ermB, but no lincosamides or erythromycin resistance.
3. Line 140-142, the MIC of the persister cells changed compared to that of planktonic cells? If the MIC changed, can it still be the persister cells?
4. The method for the “Detection of antimicrobial resistance genes” should be described in a more detailed manner. Why these antibiotic resistant genes were selected? The primers used should be given. And the DNA bands on the agarose gel should be given in the supplementary data.
5. In 3.5, why only two biological experiments were performed? In each experiment, at least three replicates should be performed.
6. In 3.6, why 8 μg/mL penicillin , 6 μg/mL clindamycin, and 4 μg/mL erythromycin were used?
Comments on the Quality of English Language
The writing of the manuscript should be checked carefully, the Latin name in the “2. Results and Discussion” should be italic. Its weared to see the writing of “0,88%” and other percent number.
Author Response
Reviewer #2 comments:
The manuscript submitted by Lannes-Costa studied the antibiotic resistance and persistence in Streptococcus agalactiae. This topic is interesting and the study is meaningful. In my opinion, there are some issues to be addressed:
- There lacks solid data for the statement of “persistence induced by biofilm-like environments” in the title. In my opinion, the present data can only supported the presence of persister cells in the biofilm-like environment.
Answer: The authors agree with the suggestion. The title was changed to “Antibiotic resistance and presence of persister cells in the biofilm-like environments in Streptococcus agalactiae”.
- In the result section, how to interpret the result that there was linB gene, mefA and ermB, but no lincosamides or erythromycin resistance.
Lines 140-148 on page 4 - We wrote the sentences to read: Interestingly, S. agalactiae strains that presented the linB, mefA, and ermB genes did not show resistance to lincosamides or erythromycin in the antimicrobial susceptibility test. Our data can be explained by the fact that specific signals or stressors are responsible for triggering the expression of resistance genes, which can lead to transient phenotypic resistance [54]. Genotypic-phenotypic correlation analysis has shown that the presence of the resistance gene may not be detected phenotypically in an unstimulated environment [55]. However, the genotypic-phenotypic correlation of antimicrobial resistance in S. agalactiae is poorly understood, requiring future studies to optimize the treatment and public health impact of invasive infections caused by this microorganism.
- Line 140-142, the MIC of the persister cells changed compared to that of planktonic cells? If the MIC changed, can it still be the persister cells?
Line 155-158 on page 4: - We wrote the sentences to read: The MIC value cannot be determined for persister cells because by progressively increasing the drug concentration, using the same high population density inoculum, the MIC value cannot be detected. In addition, when regenerated in the absence of the antimicrobial, the persisters completely restore drug susceptibility to MIC values.
- The method for the “Detection of antimicrobial resistance genes” should be described in a more detailed manner. Why these antibiotic resistant genes were selected? The primers used should be given. And the DNA bands on the agarose gel should be given in the supplementary data.
Line 299-312 on page 8: We wrote the sentences to read: The antibiotic resistance genes (ermA, ermB, mefAE, linB, tetM, and tetO; Supplemantary Materials Table S2) of S. agalactiae strains (COH1, CNCTC10/84, GBS90356, GBS1428, and GBS85147) were detected by PCR. Macrolides and lincosamides are the antibiotic alternative for patients with a history of β-lactam allergy. However, the increasing resistance among clinical strains of S. agalactiae [62] to macrolides and lincosamides, as well as the ubiquitously high (usually >80%) resistance to tetracycline in S. agalactiae [63], highlights the importance of these resistance genes. Then, the PCR amplification reaction was performed in a final 50 µL mixture volume comprised of 200 μM each dNTP, 2.5 units of GoTaq polymerase (Promega Biotechnology, Brazil), 50 ng DNA and 10 pMol of reverse and forward primers in a thermal cycler (Veriti, Applied Bio). Amplification of resistance genes was as follows: pre-denaturation at 94 °C for 5 min, denaturation at 94 °C for 1 min, annealing at 52 °C for 1 min, extension at 72 °C for 2 min and final extension ate 72 °C for 5 min. PCR products were resolved by electrophoresis in accordance with item 3.2 (Supplemantary Materials Figure S1). The primers and DNA bands on the agarose gel were included in supplementary data.
- In 3.5, why only two biological experiments were performed? In each experiment, at least three replicates should be performed.
Answer: The sentence was wrong and was corrected.
Line 318 -319 on page 8: The assay was performed in biological triplicates and repeated at two technical replicates.
- In 3.6, why 8 μg/mL penicillin, 6 μg/mL clindamycin, and 4 μg/mL erythromycin were used?
Line 332-335 on pages 8, 9 - We wrote the sentences to read: Persisters can grow at antibiotic levels well above the MIC. Thus, concentrations of 8 μg/mL penicillin (PEN; Sigma), 6 μg/mL clindamycin, (CLI; Sigma) and 4 μg/mL erythromycin (ERY; Sigma), which correspond to 16- to 267-fold higher than the MIC for each drug, were randomly selected in the present study.
Round 2
Reviewer 1 Report
Comments and Suggestions for Authors
All the issues have been addressed.
Scale bars are still hard to see, please increase font size to the same as 98% on top left corner of each image.
Reviewer 2 Report
Comments and Suggestions for Authors
The authors have made corredponding revisions and the manuscript can be accepted in the current form.